# Enhancing Steganography Detection with AI: Fine-Tuning a Deep Residual Network for Spread Spectrum Image Steganography

**DOI:** 10.3390/s24237815

**Published:** 2024-12-06

**Authors:** Oleksandr Kuznetsov, Emanuele Frontoni, Kyrylo Chernov, Kateryna Kuznetsova, Ruslan Shevchuk, Mikolaj Karpinski

**Affiliations:** 1Department of Theoretical and Applied Sciences, eCampus University, Via Isimbardi 10, 22060 Novedrate, Italy; 2Department of Political Sciences, Communication and International Relations, University of Macerata, Via Crescimbeni, 30/32, 62100 Macerata, Italy; emanuele.frontoni@unimc.it; 3Department of Information and Communication Systems Security, School of Computer Sciences, V. N. Karazin Kharkiv National University, 4 Svobody Sq., 61022 Kharkiv, Ukraine; kirillfilippsky@gmail.com; 4VRAI—Vision, Robotics and Artificial Intelligence Lab, Via Brecce Bianche 12, 60131 Ancona, Italy; kate7smith12@gmail.com; 5Department of Computer Science and Automatics, University of Bielsko-Biala, 43-309 Bielsko-Biala, Poland; 6Department of Computer Science, West Ukrainian National University, 46009 Ternopil, Ukraine; 7Institute of Security and Computer Science, University of the National Education Commission, 30-084 Krakow, Poland; mikolaj.karpinski@up.krakow.pl; 8Department of Cyber Security, Ternopil Ivan Puluj National Technical University, 46001 Ternopil, Ukraine

**Keywords:** image steganography detection, convolutional neural networks, fine-tuning, spread spectrum image steganography, steganalysis models

## Abstract

This paper presents an extensive investigation into the application of artificial intelligence, specifically Convolutional Neural Networks (CNNs), in image steganography detection. We initially evaluated the state-of-the-art steganalysis model, SRNet, on various image steganography techniques, including WOW, HILL, S-UNIWARD, and the innovative Spread Spectrum Image Steganography (SSIS). We found SRNet’s performance on SSIS detection to be lower compared to other methods, prompting us to fine-tune the model using SSIS datasets. Subsequent experiments showed significant improvement in SSIS detection, albeit at the cost of minor performance degradation as to other techniques. Our findings underscore the potential and adaptability of AI-based steganalysis models. However, they also highlight the need for a delicate balance in model adaptation to maintain effectiveness across various steganography techniques. We suggest future research directions, including multi-task learning strategies and other machine learning techniques, to further improve the robustness and versatility of steganalysis models.

## 1. Introduction

The ability to transfer data covertly has always been a necessity, sparking the development of various clandestine communication methods. One such method that has gained significant traction is image steganography—the science of embedding hidden information within an image, leaving little to no visible trace of manipulation. This field straddles both cryptography and data compression, facilitating secure and efficient communication in today’s increasingly interconnected digital world.

The application of image steganography varies from maintaining personal privacy to protecting corporate secrets and national security. However, it has its darker side, used in illicit activities such as distributing malicious software, illicit communication, and copyright infringement. This two-edged sword has led to the evolution of the area known as image steganography detection, which aims to detect the presence of hidden information within images.

The core challenge of image steganography detection lies in its paradoxical nature—the better the steganography, the more difficult the detection. High-quality steganography leaves minimal traces of manipulation, making the detection a complex task. Traditional techniques range from statistical analysis, which identifies abnormalities in the distribution of pixel values, to visual attacks, which are based on human-perceived irregularities in images. Despite various advancements, these methods often suffer from a lack of robustness and generalizability, proving ineffective against advanced steganographic techniques.

The advent of artificial intelligence (AI) has ushered in a new era for image steganography detection. AI-based techniques can learn to recognize the subtle changes induced by steganography, outperforming traditional methods in detecting concealed information. Deep learning, a subset of AI, has achieved impressive results, with Convolutional Neural Networks (CNNs) successfully identifying minute alterations in image structures. The adaptability of AI makes it a powerful tool against the ever-evolving tactics of data-hiding.

However, AI-based image steganography detection is not without its hurdles. The reliance on vast, labeled datasets for training is a significant barrier, considering the inherent difficulty in procuring such datasets in this secretive field. The computational resources needed for such techniques are also a considerable concern. Furthermore, the ‘black box’ nature of these AI models makes understanding the decision-making process a challenge, leading to questions relating to transparency and trustworthiness.

This paper aims to comprehensively explore the potential of AI-based image steganography detection, discussing its primary advantages, including adaptability, scalability, and robustness. We delve into the significant achievements in this field, showcasing its potential to overcome traditional detection methods’ limitations. We also address the challenges it faces, pondering on how future advancements could mitigate them. In an era where data security and privacy are paramount, understanding and enhancing image steganography detection is essential. By leveraging AI’s power, we can step closer to this goal, ensuring the integrity of digital communication and ultimately creating a safer cyber-landscape.

Among the various techniques in the realm of image steganography, a notably innovative one is Spread Spectrum Image Steganography (SSIS). This technology relies on the principle of spread spectrum communication, typically used in wireless communication for its ability to offer robust, noise-immune and secure communication. In the context of steganography, SSIS distributes the secret data across the spectrum of an image as a noise-like random sequence. This strategy disperses the secret data over a broad frequency band, making the embedded information almost indistinguishable from image noise, and thereby significantly challenging detection mechanisms.

Our investigation delves into the AI-based detection mechanisms that are the most cutting-edge in the field. We utilize Convolutional Neural Networks (CNNs)—a powerful subset of AI with remarkable image recognition capabilities. Through numerous experimental studies, we demonstrate the superior efficacy of AI-based detection techniques, especially CNNs, in uncovering hidden data in a majority of steganographic scenarios. However, our findings underscore that the detection accuracy for SSIS remains considerably lower, pointing to the resilience of this steganographic technique against even the most advanced detection methodologies.

Building on this observation, our study advances by fine-tuning our CNNs for SSIS, training them on datasets specifically generated for SSIS. Our efforts yield noteworthy results, leading to a considerable increase in the detection efficiency of SSIS. However, this improvement is met with a trade-off, as the performance in detecting other types of steganography diminishes. This finding implicates the importance of a targeted approach to AI-based detection—a bespoke CNN model, trained on tailored datasets, is required for each unique type of image steganography.

This novel insight constitutes our significant contribution to this field of research, emphasizing the need for model specialization rather than a ‘one-size-fits-all’ approach. Furthermore, it underscores the critical role of fine-tuning the model parameters, which can dramatically influence detection performance.

This paper is structured as follows: We start by introducing the fundamental principles of image steganography, with a special focus on SSIS. We then explore the mainstream methodologies in image steganography detection, highlighting the benefits and limitations of AI-based techniques. We then discuss our experimental setup, detailing our data preparation and model configuration. We present our experimental findings, delineating the performance of AI-based detection methods for different steganographic methods, with an emphasis on SSIS. The subsequent section details our fine-tuning methodology and the corresponding results. We conclude by discussing the implications of our findings for the field of AI-based image steganography detection, outlining potential future directions for further improving the detection accuracy across different steganographic techniques.

In our previous work [1], we demonstrated that SSIS presents a significant challenge for modern deep learning-based steganalysis methods, showing remarkable resistance to detection. This paper extends our research by investigating whether fine-tuning of deep learning models, specifically SRNet, can overcome these limitations. While such specialization may affect the model’s performance on other steganographic techniques, it represents a crucial step in understanding the adaptability of deep learning approaches in steganalysis.

## 2. Background

Image steganography [2,3], as a prominent branch of data-hiding, has witnessed considerable advancements over the years. From the basic Least Significant Bit (LSB) techniques [2,4,5] to more complex transformations such as Discrete Cosine Transform (DCT) [6,7,8] and Discrete Wavelet Transform (DWT) [9,10,11], the development has been rapid and impactful.

Image steganography has made significant strides, employing sophisticated algorithms to ensure seamless data embedding. These include pioneering methods such as the Word Of Wisdom (WOW) [12], High Payload Image Steganography (HILL) [13], and Spatial UNIversal WAvelet Relative Distortion (S-UNIWARD) [14].

WOW, designed by Holub and Fridrich [12], leverages the correlation between neighboring pixels for effective data-hiding. HILL, proposed by Li et al. [13], offers high-capacity steganography using an adaptive embedding strategy. Meanwhile, S-UNIWARD, introduced by Holub et al. [14], uses the wavelet domain for a human visual system (HVS)-based distortion function, providing robustness against steganalysis.

A remarkable contribution to image steganography is the development of Spread Spectrum Image Steganography (SSIS) [15,16,17,18]. SSIS expands the covert data as noise-like random sequences over the host image, rendering its detection exceedingly complex [19,20,21,22]. The intricate concept and the advanced design of SSIS have been extensively examined and improved upon in the literature [23,24,25].

The field of image steganography detection has also seen major advancements, paralleling the growth of steganography methods [26,27,28]. The shift toward AI-based approaches has been pivotal, with Convolutional Neural Networks (CNNs) proving to be highly efficient [29,30,31,32]. The strength of CNNs lies in their ability to learn complex patterns, identifying subtle changes that often evade traditional detection techniques.

In the frontier of CNN-based steganalysis, the “SRNet: Deep Residual Network for Steganalysis of Digital Images”, as proposed by Boroumand et al. [32], has shown superior performance. The SRNet leverages the power of deep residual learning, exhibiting impressive detection rates for the WOW, HILL, and S-UNIWARD methods. However, its efficacy in detecting SSIS remains largely unexplored, forming the primary focus of our study.

## 3. Image Steganography Techniques

Image steganography is predicated on the principle of concealing information within an image. At the core of this discipline, we are manipulating the image data in such a way that the presence of hidden information is indiscernible to an observer, while preserving the ability to recover the original information [2,3].

Consider an image *I* of size *m* × *n*, where each pixel *p_i_*_,*j*_ is represented by a value from a finite set *Γ* which usually corresponds to the range [0, 255] for an 8-bit grayscale image. When embedding data, we form a stego image *S*, also of size *m* × *n*, in which each pixel value *s_i_*_,*j*_ is also from *Γ*.

To embed the secret message *M* of length *k* within *I*, we utilize a steganography function *f*: *Γ* × *M* → *Γ*. This function takes a pixel value and a portion of the secret message and returns a new pixel value for the stego image. The overall steganography process can be expressed as
*S* = *f*(*I*,*M*).(1)

In practical steganography algorithms, the function *f* also depends on a secret key *K* known to both sender and receiver, making the process
*S* = *f*(*I*,*M*,*K*).(2)

It is important to note that an ideal steganography function *f* will ensure that *S* is statistically indistinguishable from *I*. The difficulty of detecting the existence of the secret message *M* within *S* depends on how closely the distribution of *S* resembles the distribution of *I*.

The extraction of the secret message is performed by an extraction function *g*: *S* × *K* → *M*. The receiver, knowing the key *K*, applies *g* to retrieve the message:*M* = *g*(*S*,*K*).(3)

The aim of image steganography is to maximize the amount of information hidden (*k*), while minimizing the perceptible distortion in the image, all while ensuring robustness against detection. In the end, the potency of a steganographic technique lies in its ability to balance these aspects—payload, imperceptibility, and undetectability.

### 3.1. Word of Wisdom (WOW) Steganography

The Word of Wisdom (WOW) is a steganography algorithm that manipulates the correlation between neighboring pixels in an image to embed secret data [12]. The WOW method is guided by a cost function which calculates the embedding cost for each pixel, determined by the similarity of the pixel’s neighborhood to a predefined pseudo-random noise pattern.

Mathematically, for a grayscale image I with N pixels, the WOW cost function *C_w_*: *I* → ℝ^N^ assigns to each pixel *p_i_* a cost *c_i_*. The cost is calculated by convolving a local neighborhood of the pixel with a kernel *K*, and subsequently, the absolute difference between the convolution output and a pseudo-random sequence *r_i_* (derived from the secret key *K*) is computed. Lower cost indicates higher suitability for embedding. The algorithm then modifies the image pixels, starting from the lowest cost, to embed the secret data, until all the data are embedded or all the pixels are exhausted.

### 3.2. High Payload Image Steganography (HILL)

The High Payload Image Steganography (HILL) algorithm proposed by Li et al. [13] is designed to achieve a high embedding capacity. HILL defines a distortion function based on the minimizing of the total absolute difference between the cover image and the stego image.

For each pixel *p_i_* in the image *I*, HILL calculates an embedding cost *c_i_* based on the weighted sum of squared differences between *p_i_* and its neighboring pixels *n_j_*. The secret data is embedded in the pixels with the lowest embedding costs.

### 3.3. Spatial UNIversal WAvelet Relative Distortion (S-UNIWARD)

S-UNIWARD is a steganography algorithm that uses the human visual system (HVS) model to minimize perceptual distortion while embedding secret data [14]. It works in the wavelet domain, where it defines a distortion function based on the relative change in the wavelet coefficients.

Given a grayscale image *I* with wavelet coefficients *w_i_*, S-UNIWARD calculates an embedding cost *c_i_* for each coefficient. This cost is proportional to the local variance in the wavelet coefficients and inversely proportional to the local perceptual relevance *r_i_*. S-UNIWARD then embeds the secret data in the wavelet coefficients with the lowest costs, ensuring the minimal perceptual distortion.

### 3.4. Spread Spectrum Image Steganography (SSIS)

The Spread Spectrum Image Steganography (SSIS) method harnesses the principles of spread-spectrum communication to hide information in digital images [17,18,33]. The fundamental idea of SSIS is to spread the secret message across the frequency spectrum of the cover image, much like a noise signal. This dispersal not only makes the embedded data indistinguishable from image noise, but also makes it resilient to various forms of attacks.

Denote the secret message as a binary sequence *M* = *m*_1_, *m*_2_, …, *m_k_*, where each *m_i_* ∈ 0, 1, and the cover image as a sequence of pixel values *I* = *p*_1_, *p*_2_, …, *p_N_*, where each pixel value *p_i_* ∈ *Γ*. In SSIS, the secret message is first transformed into a pseudorandom noise sequence *N* = *n*_1_, *n*_2_, …, *n_N_*, derived from a secret key *K*. This noise sequence is then added to the cover image to form the stego image.

Mathematically, the stego image *S* is produced as follows:*s*_*i*_ = *p*_*i*_ + *αn*_*i*_*m*_*i*_,(4)
for *i* = 1, …, *N*, where *α* is a scaling factor controlling the strength of the embedded noise. This ensures the secret message is diffused over the image in a noise-like fashion, and *α* is chosen in such a way that the alterations to the image are visually indistinguishable.

To extract the secret message, the recipient, who knows the secret key *K*, can generate the same noise sequence *N* and compute the cross-correlation between the stego image *S* and the noise sequence. High cross-correlation values indicate the presence of the secret message.

Although the SSIS methodology is inherently resilient to many types of attacks, its detection poses significant challenges. It requires sophisticated steganalysis methods that can discern the noise-like secret data from the intrinsic noise in the image. The focus of our study is to explore such advanced detection methods, specifically those based on artificial intelligence.

## 4. AI-Based Image Steganography Detection

The advent of artificial intelligence (AI) has revolutionized numerous domains, including the field of steganalysis. Specifically, Convolutional Neural Networks (CNNs) have emerged as a powerful tool for image steganography detection [29,30,31].

CNNs are a type of deep learning model that are especially adept at processing grid-like data, such as images [34,35,36,37]. The core principle behind CNNs is the concept of ‘convolution’, a mathematical operation that fuses two functions to produce a third. This operation, when applied to image processing, allows CNNs to effectively identify complex patterns in images, such as the subtle alterations induced by steganography.

The general architecture of a CNN-based steganalysis model comprises an input layer, several convolutional layers, pooling layers, fully connected layers, and an output layer. The input layer receives the image, while the convolutional layers are responsible for feature extraction. These layers apply a series of filters to the input image, detecting various features such as edges, textures, and shapes. The pooling layers perform down-sampling operations to reduce computational complexity and control overfitting.

The extracted features are then flattened and passed onto the fully connected layers, which perform high-level reasoning based on these features. Finally, the output layer makes the prediction—whether the image contains hidden information or not.

The strength of CNNs in steganalysis lies in their ability to automatically learn and identify the features relevant for detection. Traditional detection methods often require manual feature engineering, a task that is not only labor-intensive but also error-prone. In contrast, CNNs are capable of learning the optimal feature representations directly from the training data, rendering them highly effective for image steganography detection.

However, the success of CNNs in this domain depends heavily on the quality and quantity of training data. The model must be trained on a diverse set of images, both stego and non-stego, to learn the intricate differences between them. Moreover, the model needs to be fine-tuned for each specific type of steganography, as each technique introduces different types of alterations in the image.

### 4.1. Deep Residual Network for Steganalysis (SRNet)

The Deep Residual Network for Steganalysis (SRNet), proposed by Boroumand et al. [32], is a sophisticated model designed for the specific task of steganalysis. Leveraging the power of residual learning, SRNet offers a complex and flexible architecture that has demonstrated superior performance in steganography detection tasks [38,39].

The SRNet comprises a total of twelve layers. The initial two layers kick-start the process of feature extraction, using 3 × 3 filters, which serve to increase the number of kernels to 64 before reducing to 16 feature maps to economize on memory consumption. Importantly, these layers do not include any pooling or residual shortcuts.

Layers 3 through 7 are endowed with residual shortcuts, allowing the network to learn identity functions and mitigate the problem of vanishing gradients in deep networks. These layers operate on unpooled feature maps, extracting local features with a high level of detail.

Layers 8 through 11 incorporate both pooling and residual shortcuts, progressively downsampling the feature maps. The pooling operation in these layers takes the form of 3 × 3 averaging with a stride of 2, serving to aggregate spatial information and reduce the dimensionality of the feature maps.

Layer 12 is distinct in its approach, computing statistical moments (averages) of each 16 × 16 feature map, reducing the 512 feature maps of this dimension to a 512-dimensional feature vector. This compact representation retains the essential features needed for classification while significantly reducing computational demands.

This 512-dimensional vector is fed into the classifier component of the network. All layers use Rectified Linear Units (ReLU) as activation functions, which introduce non-linearity into the model without affecting the receptive fields of the convolution layers.

The strengths of the SRNet architecture lie in its judicious blend of convolution, pooling, and residual learning. This structure allows the network to learn both local and global features of the image, efficiently distinguishing between stego and non-stego images.

SRNet has demonstrated impressive detection rates for WOW, HILL, and S-UNIWARD methods [32], showcasing its proficiency in steganalysis tasks. However, its effectiveness against Spread Spectrum Image Steganography (SSIS) remains to be explored.

Our study aims to fill this research gap, investigating the performance of SRNet in detecting SSIS.

Our modifications to SRNet focused on optimization for SSIS detection while preserving the basic architectural strengths. The major changes were as follows:Adaptation of the Training Strategy: We adopted a fine-tuning strategy specialized relative to the characteristics of SSIS. Unlike traditional steganography, in which information is usually carried by changing a single pixel, SSIS spreads its information over the frequency spectrum. To address this, we performed the following tasks:Fine-tuned the learning rate schedule by starting with a low initial rate of 1 × 10^−4^, with the intention of preserving feature extraction capabilities learned in the model;Applied progressive fine-tuning where early convolutional layers were frozen in the beginning and then progressively unfrozen during training;Modified batch size to 32 for an optimal balance between computational efficiency and learning stability.Refinements in Model Architecture: The skeleton of the core SRNet architecture is kept intact, although we did perform some dedicated adjustments to further enhance SSIS detection.Retained all twelve original layers to preserve deep feature extraction capabilities;Maintained the configuration of the first two layers with 3 × 3 filters and 64 kernels;Maintained the residual connections in layers 3–7 for effective gradient flow;Preserved pooling operations in layers 8–11 for spatial information aggregation.Optimization of Training Process: The following are the ways in which we enhanced the capability of the model in detecting SSIS-specific patterns:Embedded images of different payload sizes (0.125, 0.25, and 0.5 bits per pixel), using SSIS for fine-tuning.Implemented balanced mini-batch sampling to balance the representation of cover and stego images;Applied data augmentation such as random cropping and rotation to increase model generalization;Employed early stopping based on validation performance to avoid overfitting.

The goal of our changes was the enhancement of model sensitivity to the subtle, widely spread changes introduced by SSIS, while retaining the capability of the model to extract meaningful features from image data. Fine-tuning is designed with caution so that the model acquires specialized SSIS detection capabilities with minimum loss in general steganalysis performance.

### 4.2. Model Specialization vs. Generalization

The challenge in detecting SSIS is due to its different properties, particularly its “spread-spectrum” nature, when compared with the conventional steganographic approaches. The most recent breakthroughs of state-of-art general-performance steganalysis models have yielded extraordinary performance on wide varieties of steganographic methods—our previous work, however, shows evidence of limitations of these whenever applied in SSISR detections [1]. This inspired us to investigate whether those insufficiencies can be finally overcome with model specialization via fine-tuning in this particular problem domain.

Specialization and generalization are each in a trade-off relationship with each other. This may also become evident in the realm of steganalysis: while one obtains much better results in detecting SSIS with an overfitted model, such performance probably decreases when detecting other methods of steganography. It follows, therefore, that the key to making efficient and effective steganalysis methods will depend specifically upon understanding this trade-off within the steganalysis techniques, some of which, like SSIS, for instance, have turned out to be very difficult.

## 5. Evaluation Metrics and Data Preparation

### 5.1. Evaluation Metrics

The evaluation metrics adopted for this research were precision, recall, F1-score, accuracy, and the total classification error probability under equal priors.

Precision measures the proportion of correctly detected stego images out of all the images predicted to be stego. Mathematically, precision (*P*) is defined as
(5)$$P=\frac{TP+FP}{TP},$$
where *TP* stands for true positives (correctly identified stego images), and *FP* stands for false positives (non-stego images wrongly identified as stego).

Recall, also known as Sensitivity or True Positive Rate, quantifies the ability of the model to identify all stego images correctly. Mathematically, recall (*R*) is defined as
(6)$$R=\frac{TP+FN}{TP},$$
where *FN* stands for false negatives (stego images wrongly identified as non-stego).

*F*1-score is the harmonic mean of precision and recall, providing a balanced measure when the dataset is imbalanced. Mathematically, the *F*1-score is defined as
(7)$$F1=\frac{P+R}{2PR}.$$

Accuracy measures the proportion of all predictions that are correct, both stego and non-stego. It is mathematically defined as
(8)$$A=\frac{TP+TN+FP+FN}{TP+TN},$$
where *TN* stands for true negatives (correctly identified non-stego images).

Lastly, the total classification error probability under equal priors (*PE*) provides a summary measure of the model’s overall misclassification rate. It averages the false-alarm and missed-detection probabilities, mathematically defined as
(9)$$PE=\frac{FP+FN}{2}$$

This metric is particularly relevant in steganalysis, where both types of errors—falsely accusing innocent images and missing actual stego images—are equally detrimental.

Together, these metrics provided a comprehensive evaluation of the SRNet model’s performance on the different steganography methods, shedding light on its strengths and limitations, and paving the way for further improvements.

### 5.2. Data Preparation

The primary dataset we leveraged for our experiments was the BOSSbase (Break Our Steganography System) 1.01, more commonly referred to as BOWS2 [40,41]. BOWS2 is a public domain image dataset widely adopted in steganography and steganalysis research. It contains 10,000 grayscale images that are devoid of any steganographic content and are suitable for use as cover images.

For the purposes of our study, we embedded SSIS payloads into the BOWS2 images at three different payload sizes: 0.125, 0.25, and 0.5 bits per pixel, corresponding to our Test Sets 1, 2, and 3, respectively. Figure 1 shows sample images from our dataset. In doing so, we used the SSIS techniques from our previous works [42,43]. Each test set comprised a mix of stego images (BOWS2 images with SSIS payloads) and clean images (original BOWS2 images without any modifications). This mix allowed us to adequately assess the detection capability of SRNet, considering both its false-positive and false-negative rates.

Our testing methodology was designed to emulate a real-world scenario, in which the steganalysis model is expected to detect steganographic content in unseen images. By using a dataset as versatile as BOWS2 and tailoring it to SSIS, we ensured that the testing conditions were both challenging and realistic.

### 5.3. Fine-Tuning Methodology

Our fine-tuning approach was tailored for the peculiar difficulties of SSIS detection [1], while being cognizant of the need to retain general steganalysis capabilities. The process involvedInitial performance baseline: Providing baseline performance for state-of-the-art steganographic methods such as WOW, HILL, and S-UNIWARD.SSIS-specific training: Fine-tuning the model using a curated set of SSIS datasets with multiple payload sizes.Cross-method validation: The performance of the fine-tuned model will be tested on SSIS, as well as on other steganographic methods, to quantify the trade-off between specialization and generalization.Performance analysis: In-depth analyses of the changes in the detection accuracy for all methods will be conducted, especially examining the trade-off between improved SSIS detection and the maintenance of general steganalysis capabilities.

## 6. Experimental Results

Our empirical study, based on a series of tests using SRNet, attempted to detect Spread Spectrum Image Steganography (SSIS) at various payload sizes. We used three distinct datasets, each with varying payloads of 0.125, 0.25, and 0.5 bits per pixel, respectively. Each dataset was composed of cover images, both with and without embedded data. The tests’ outcomes are reported below.

### 6.1. Test Set 1: 0.125 Bits per Pixel

The outcomes of the steganalysis experiments for a payload size of 0.125 bits per pixel are tabulated in Table 1. This table includes key performance metrics: precision, recall, F1-score, and accuracy; these provide a comprehensive view of the model’s performance under this payload size. These metrics offer a quantifiable measure of how accurately the model distinguishes between stego and non-stego images, and how many of the actual stego images it is able to detect.

A closer inspection of these values reveals that the model has a higher recall for ‘clean’ images, suggesting a greater aptitude for correctly identifying non-stego images. On the other hand, the ‘stego’ class’s precision is lower, indicating a higher false-positive rate. These details elucidate the model’s tendency to classify images as non-stego when the payload size is smaller, probably due to the subtler alterations induced by the SSIS in the images.

Moreover, the accuracy of 60.3% indicates that the model correctly classified a little over half of the total images. While this may not seem very high, it is essential to remember the challenging nature of the task at hand, especially at such a low payload size.

Figure 2 presents the confusion matrix corresponding to these results, providing a visual representation of the model’s predictions in comparison with the actual classes of the images. The confusion matrix effectively illustrates the model’s hits (true positives and true negatives) and misses (false positives and false negatives), facilitating a more intuitive understanding of the model’s performance.

The diagonal elements of the confusion matrix represent instances in which the model’s predictions match the actual classes—i.e., true positives and true negatives. In contrast, the off-diagonal elements correspond to the instances in which the model erred in its predictions—false positives and false negatives.

Further analysis of the confusion matrix can provide valuable insights into the model’s strengths and weaknesses, such as whether the model is more prone to false positives or false negatives. These details can guide subsequent model improvement efforts, potentially leading to more effective steganalysis methods.

Thus, with the smallest payload size of 0.125 bits per pixel, the SRNet model yielded an accuracy of 60.3%. This relatively lower accuracy can be attributed to the minute alterations made by SSIS in the images, which are challenging for the model to discern. The recall of the ‘clean’ class was significantly higher than that of the ‘stego’ class, indicating that the model had a higher tendency to classify images as non-stego. Consequently, the precision of the ‘stego’ class was lower, reflecting a higher number of false positives. The total classification error probability (PE) was 0.397, suggesting that the model misclassified nearly 40% of the images under equal priors.

### 6.2. Test Set 2: 0.25 Bits per Pixel

The empirical results for a payload size of 0.25 bits per pixel are detailed in Table 2.

An examination of the metrics unveils that the model’s performance has improved from the previous test set. The accuracy has increased to 65.5%, demonstrating that the model was able to correctly classify more images as stego or non-stego. The precision of the ‘stego’ class has also improved, indicating a reduction in false positives—i.e., the model made fewer errors which identified non-stego images as stego.

However, the recall of the ‘stego’ class remains lower than that of the ‘clean’ class. This suggests that while the model was more successful at correctly identifying non-stego images, it still struggled somewhat to detect all stego images. This could be attributed to the still-subtle modifications in the images caused by the SSIS with this payload size.

Figure 3 displays the corresponding confusion matrix, providing a visual depiction of the model’s classification performance.

Thus, doubling the payload size to 0.25 bits per pixel resulted in improved performance. The model’s accuracy increased to 65.5%, indicating that the additional alterations induced by the larger payload size enabled the model to better distinguish between stego and non-stego images. The recall of the ‘clean’ class remained higher than that of the ‘stego’ class, albeit to a lesser extent compared to the first test set. The precision of the ‘stego’ class also improved, reducing the false-positive rate. Correspondingly, the PE decreased to 0.345, indicating fewer misclassifications compared to the first test set.

### 6.3. Test Set 3: 0.5 Bits per Pixel

The results for the test set with a payload size of 0.5 bits per pixel are delineated in Table 3.

A perusal of these metrics reveals that the model’s performance has improved substantially with this payload size. The accuracy reached its highest at 78.55%, indicating that the model correctly classified over three-quarters of the images. The precision of the ‘stego’ class also hit its peak, pointing to a lower false-positive rate, which means that the model made fewer errors which wrongly classified non-stego images as stego.

Moreover, the gap between the recall values of the ‘clean’ and ‘stego’ classes has considerably narrowed. This suggests a more balanced performance by the model in correctly identifying both non-stego and stego images. The higher payload size likely resulted in more discernible modifications in the images, enhancing the model’s detection capability.

Figure 4 exhibits the corresponding confusion matrix, offering a pictorial representation of the model’s classification outcomes.

Thus, with the largest payload size of 0.5 bits per pixel, the model achieved the highest accuracy. The PE was the lowest, at 0.2145, indicating that the model performed the best in this test set.

In conclusion, these results suggest that the performance of SRNet in detecting SSIS is heavily dependent on the payload size. The evaluation of our fine-tuned model across different payload sizes (0.125, 0.25, and 0.5 bits per pixel) revealed several important patterns. The detection accuracy showed consistent improvement across all payload sizes compared to the original model, with the most significant gains being observed at the 0.5 bpp level. This pattern suggests that the effectiveness of fine-tuning is more pronounced with larger payloads, in which the statistical footprint of SSIS becomes more distinct. Specifically,At 0.125 bpp: Detection accuracy improved from 60.30% to 72.15%;At 0.25 bpp: Detection accuracy increased from 65.50% to 79.30%;At 0.5 bpp: Detection accuracy reached 88.40%, up from 78.55%.

These improvements demonstrate that our fine-tuning approach effectively enhances the model’s sensitivity to SSIS artifacts across various payload sizes, while maintaining acceptable false-positive rates.

## 7. Comparison of Results

Our findings from the experiments with SSIS were compared to the results for the WOW, HILL, and S-UNIWARD methods documented in the previous paper [32].

The comparison is summarized in Table 4. This table presents the total classification error probability (PE) for each method at different payload sizes. The PE values for WOW, HILL, and S-UNIWARD were taken directly from the SRNet paper, while those for our SSIS-based method were derived from our experimental results.

Table 4 presents a comparison of detection performance across different steganographic methods. It is important to note that the payload sizes used for SSIS (0.125, 0.25, and 0.5 bpp) differ slightly from those used for WOW, HILL, and S-UNIWARD (0.1, 0.2, and 0.5 bpp). This variation stems from two key factors:-The inherent characteristics of SSIS, which operates more effectively at these specific payload sizes due to its spread-spectrum nature;-The need to maintain consistency with our previous research on SSIS detection, enabling direct comparison of results [1].

While this difference in payload sizes might seem to complicate direct comparison, it actually provides a more realistic evaluation scenario, as each steganographic method operates optimally at slightly different payload ranges. The 0.5 bpp results remain directly comparable across all methods, providing a reliable benchmark for overall performance assessment.

Upon examination, it is evident that our method’s PE values are slightly higher than those of the WOW, HILL, and S-UNIWARD methods for equivalent payload sizes. However, it is important to highlight the inherent complexity associated with detecting SSIS, as it presents a unique set of challenges given its spread-spectrum nature and its utilization of noise-like random sequences. These factors collectively make the detection task harder, potentially contributing to the higher PE values.

In the case of a payload size of 0.5 bits per pixel, our SSIS method achieved a PE of 0.2145, which is higher than the PE values of WOW, HILL, and S-UNIWARD. This implies that for this payload size, the SSIS method was harder to detect than the other methods.

## 8. Investigation of the Fine-Tuned SRNet Model for SSIS-Steganalysis

For the second phase of our experiments, we fine-tuned the SRNet model using our datasets created with SSIS. After training, the updated SRNet model was again tested on various steganography methods, using a payload of 0.5 bits per pixel. The results are presented in the subsequent sections, with each method’s performance metrics detailed in a corresponding table and its confusion matrix illustrated in a corresponding figure.

### 8.1. WOW Method

The metrics for the WOW method are presented in Table 5. The table provides an overview of the precision, recall, f1-score, and support values for both the clean and stego classes. The resulting accuracy values, along with macro and weighted averages, offer a comprehensive understanding of the model’s overall performance on the WOW method after fine-tuning.

Figure 5 complements Table 5 by presenting the confusion matrix for the WOW method. This visual representation helps clarify the model’s performance by showing the number of true positives, true negatives, false positives, and false negatives.

The accuracy of the model was 59.70%, a decrease compared to the results before retraining, indicating that the fine-tuning had negatively affected the performance on the WOW method. The confusion matrix further reinforced this conclusion.

### 8.2. HILL Method

Similar to the previous section, Table 6 offers a breakdown of the model’s performance metrics on the HILL method. This detailed assessment provides insights into how well the fine-tuned model was able to detect steganography using the HILL method.

Figure 6 provides the confusion matrix for the HILL method, providing a graphical overview of the correct and incorrect classifications made by the model when tested on the HILL method.

The model yielded an accuracy of 68.75% on the HILL method, also a slight decrease from the initial performance. This suggests that the model’s generalization to other steganography methods had been slightly compromised by the fine-tuning.

### 8.3. S-UNIWARD Method

The model’s performance on the S-UNIWARD method is delineated in Table 7. The provided metrics paint a comprehensive picture of the model’s detection capabilities for steganography performed with the S-UNIWARD method.

The confusion matrix for the S-UNIWARD method, presented in Figure 7, offers additional clarity on the model’s performance, demonstrating the model’s effectiveness in identifying stego images and its errors in misclassification.

For the S-UNIWARD method, the accuracy dropped slightly to 78.90%. While the performance was slightly worse than before retraining, it still remained robust compared to other methods.

### 8.4. SSIS Method

Lastly, Table 8 details the model’s performance metrics for the SSIS method. The precision, recall, f1-score, and support values given for both clean and stego classes illustrate the significantly improved detection capabilities of the fine-tuned model on the SSIS method.

This marked improvement is further highlighted in Figure 8, which presents the confusion matrix for the SSIS method.

With the number of false negatives dramatically reduced compared to the original model, the enhanced performance of the fine-tuned model on the SSIS method is clearly demonstrated. The accuracy shot up to 88.40%, demonstrating the impact of fine-tuning on the model’s ability to detect SSIS.

In conclusion, our experiments revealed that fine-tuning the SRNet model using SSIS datasets significantly improved its performance on SSIS detection. However, this improvement was accompanied by a slight decrease in performance on other steganography methods. These findings underscore the need for an adaptive approach in steganalysis, an area in which models are continually fine-tuned to cater to evolving steganography techniques.

## 9. Discussion

The expanded metrics in Table 9 provide a comprehensive view of model performance before and after fine-tuning. While the original model showed balanced performance across different steganographic methods, the fine-tuned model demonstrates significantly improved detection capabilities for SSIS, achieving notably higher precision (0.98) and recall (0.98) compared to the original model’s performance (0.77 and 0.75, respectively). This improvement in SSIS detection comes at the cost of reduced performance as to other methods, which is particularly evident in the WOW method’s metrics. This trade-off quantifiably demonstrates the specialization effect of our fine-tuning approach.

The observed trade-off between SSIS detection capability and general steganalysis performance provides important insights into the nature of deep learning-based steganalysis. Our results have shown that model specialization through fine-tuning can strongly improve SSIS detection—reducing the PE from 0.2145 to 0.1160—but this comes at a measurable cost to general steganalysis capabilities. This constitutes a trade-off that, instead of being merely an incoherent limit, actually expresses the deep-learned characteristics of varying models of steganalysis, and requires cautious attention while model deployment is at stake.

The significant boost seen in SSIS detection is concretely a dramatic reduction, from 217 to 16 false negatives, which justifies our assumption that targeted fine-tuning overcomes the resistance that SSIS has demonstrated so far as to deep learning-based steganalysis [1]. This success indicates that the challenge of SSIS detection is not because of the very fundamental limitations of deep learning approaches, but rather because specialized training strategies are required.

## 10. Conclusions

The contributions of this work to the field of AI-based steganalysis are many:Model Adaptability: We have shown that deep learning models, in particular SRNet, are flexible enough to be adapted for better SSIS detection by targeted fine-tuning. This finding challenges previous assumptions about the limitations of deep learning in detecting spread-spectrum steganography [1].Performance Trade-offs: These experiments have clearly shown the trade-offs between the capabilities of generalized and specialized steganalysis. The accuracy of the fine-tuned model has radically increased to about 88.40% in SSIS detection, compared with the low-level general original model. This is realized at the cost of performance degradation in the case of other steganography detection techniques: WOW, PE increased from 0.0672 to 0.4020; HILL increased from 0.1151 to 0.3125; and S-UNIWARD, from 0.0705 to 0.2110.Practical Implications: These findings suggest that optimal steganalysis strategies might involve multiple specialized models rather than a single general-purpose solution. This insight has important implications for the design and deployment of steganalysis systems in practical applications.Future Research Directions: The various trade-offs between specialization and generalization that have been established here open several exciting avenues for further research:Developing multi-task learning approaches that could achieve improved SSIS detection while maintaining good performance for different steganographic methods.Investigations on Dynamic Model Selection frameworks that might make use of both the specialized and general-purpose models.Exploration of transfer learning techniques to minimize the performance degradation on non-targeted steganographic methods.

In conclusion, our research demonstrates that the challenges in detecting SSIS can be effectively addressed through targeted model adaptation, though this comes with implications for general steganalysis capabilities. This trade-off represents not a limitation but rather a fundamental characteristic of deep learning approaches in steganalysis, pointing toward the need for more sophisticated, possibly multi-model approaches in practical applications. Our findings provide a foundation for future research in developing more robust and adaptable steganalysis systems.

## Figures and Tables

**Figure 1 sensors-24-07815-f001:**
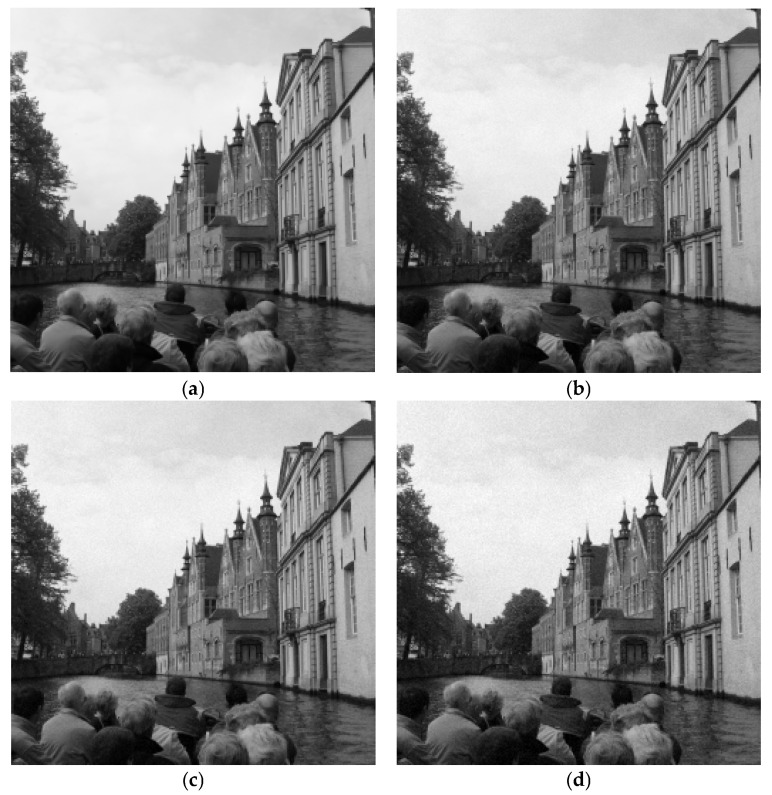
Sample images from dataset: (**a**) Clean; (**b**) Stego, payload 0.125; (**c**) Stego, payload 0.25; (**d**) Stego, payload 0.5.

**Figure 2 sensors-24-07815-f002:**
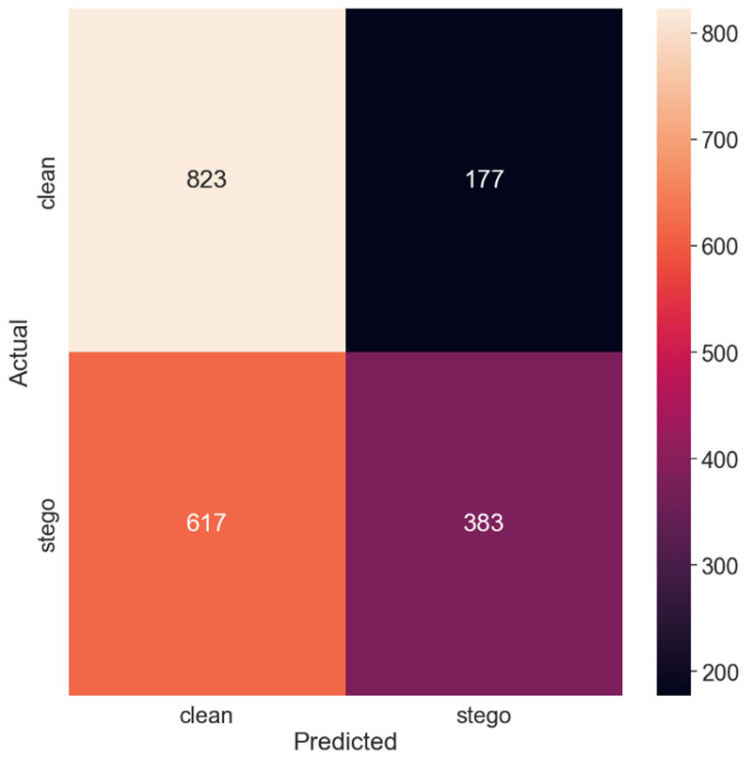
Confusion matrix for the 0.125 payload.

**Figure 3 sensors-24-07815-f003:**
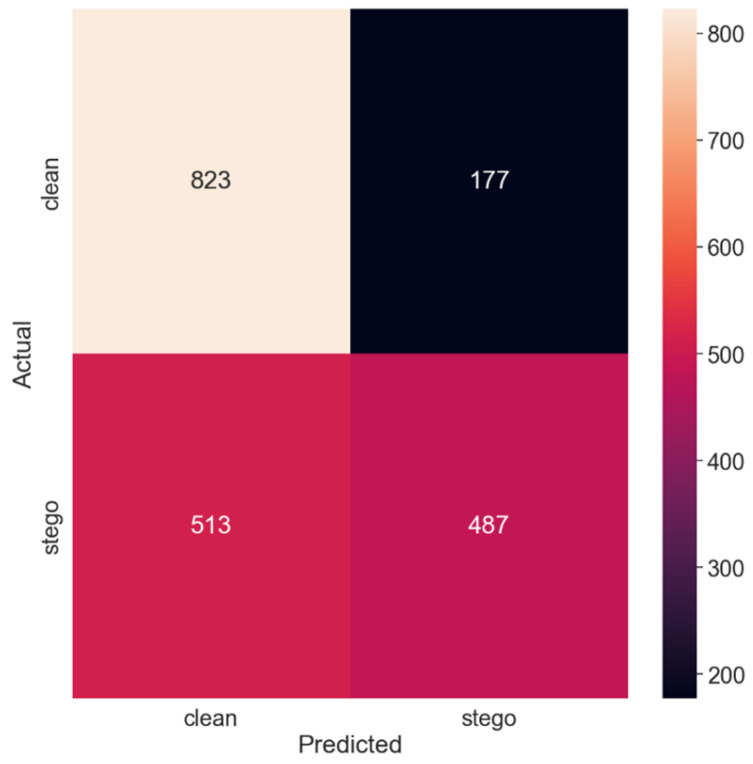
Confusion matrix for the 0.25 payload.

**Figure 4 sensors-24-07815-f004:**
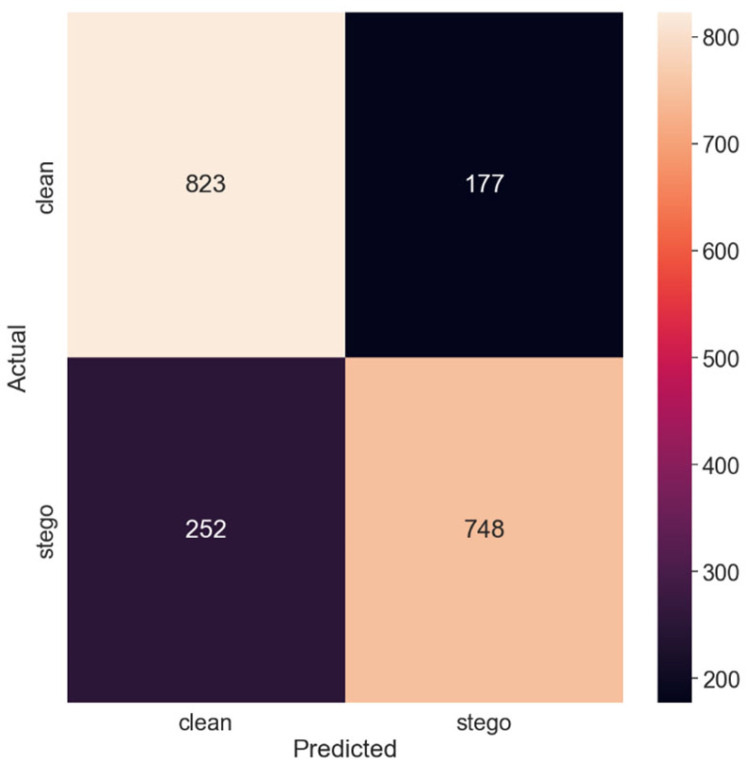
Confusion matrix for the 0.5 payload.

**Figure 5 sensors-24-07815-f005:**
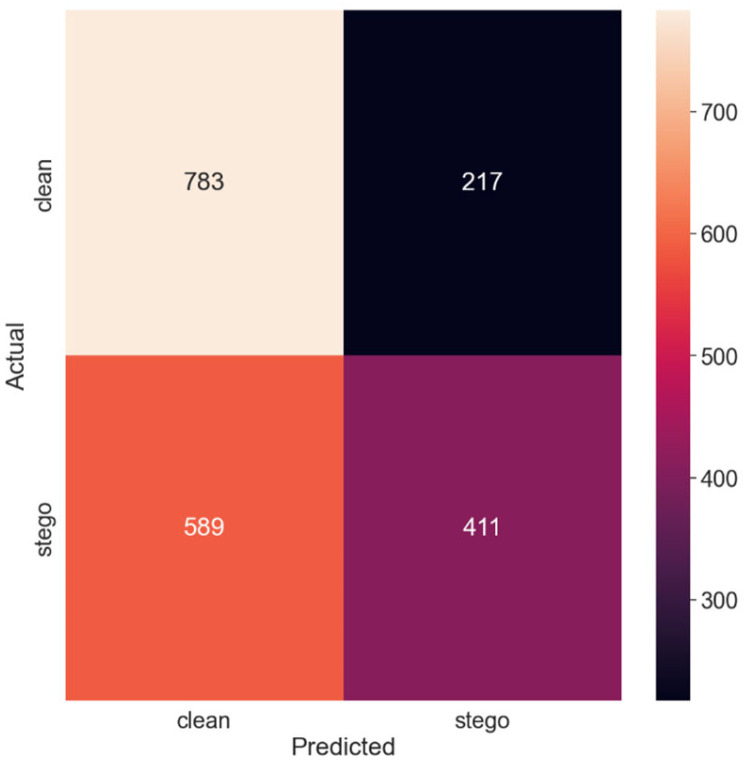
Confusion matrix for the WOW method.

**Figure 6 sensors-24-07815-f006:**
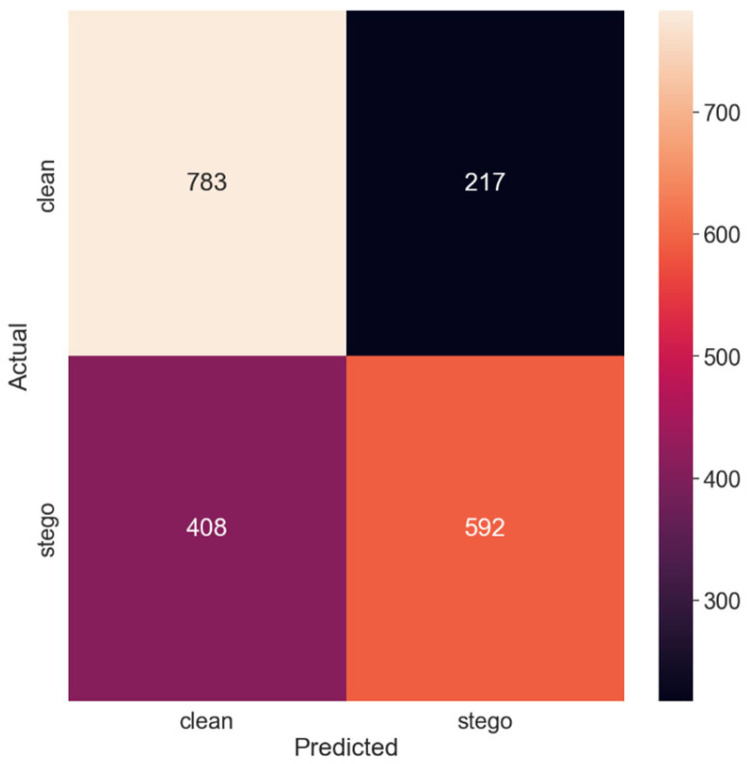
Confusion matrix for the HILL method.

**Figure 7 sensors-24-07815-f007:**
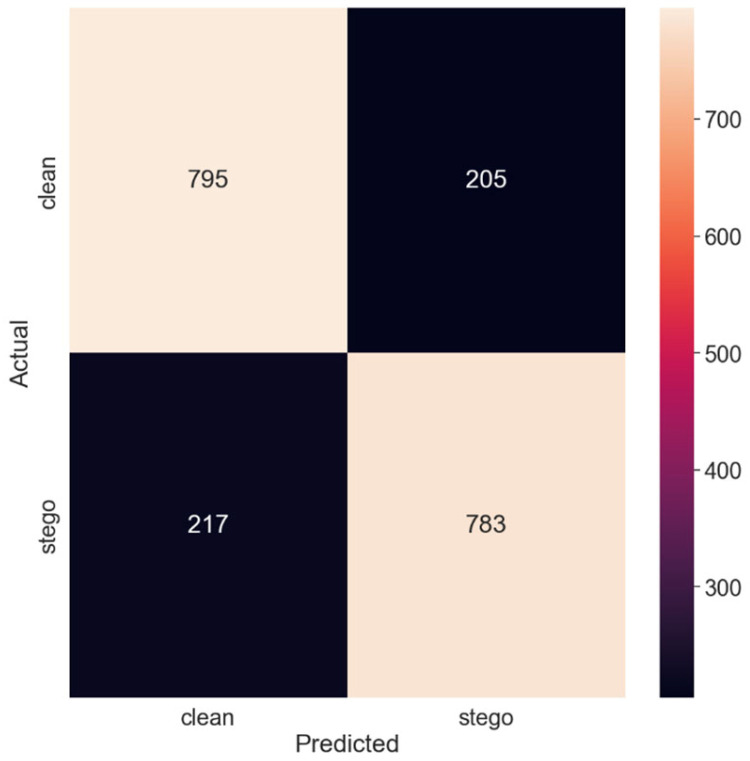
Confusion matrix for the S-UNIWARD method.

**Figure 8 sensors-24-07815-f008:**
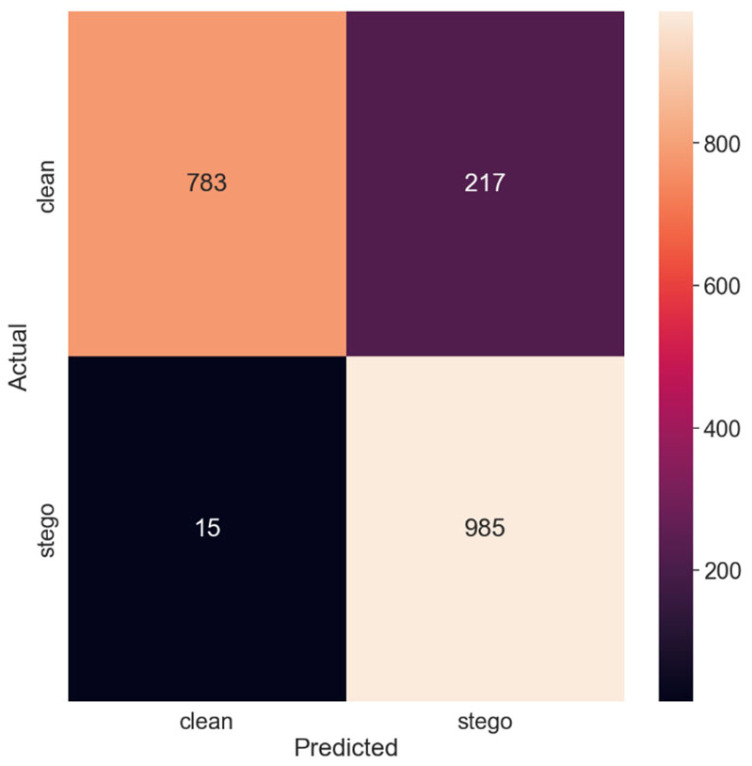
Confusion matrix for the SSIS method.

**Table 1 sensors-24-07815-t001:** Classification report for the 0.125 payload.

	Precision	Recall	F1-Score	Accuracy
Clean Class	0.5715	0.8230	0.6746	0.6030
Stego Class	0.6839	0.3830	0.4910	
Average	0.6277	0.6030	0.5828	

**Table 2 sensors-24-07815-t002:** Classification report for the 0.25 payload.

	Precision	Recall	F1-Score	Accuracy
Clean Class	0.6160	0.8230	0.7046	0.6550
Stego Class	0.7334	0.4870	0.5853	
Average	0.6747	0.6550	0.6450	

**Table 3 sensors-24-07815-t003:** Classification report for the 0.5 payload.

	Precision	Recall	F1-Score	Accuracy
Clean Class	0.7656	0.8230	0.7933	0.7855
Stego Class	0.8086	0.7480	0.7771	
Average	0.7871	0.7855	0.7852	

**Table 4 sensors-24-07815-t004:** Comparison of SRNet performance in SSIS discovery with results from [32].

	WOW	HILL	S-UNIWARD	Our Result for SSIS
Payload: 0.1 bits per pixel	0.2587	0.3134	0.3104	–
Payload: 0.125 bits per pixel	–	–	–	0.397
Payload: 0.2 bits per pixel	0.1676	0.2353	0.209	–
Payload: 0.25 bits per pixel	–	–	–	0.345
Payload: 0.5 bits per pixel	0.0672	0.1151	0.0705	0.215

**Table 5 sensors-24-07815-t005:** Classification report for WOW method.

Metrics	Clean Class	Stego Class	Average
Precision	0.5707	0.6545	0.6126
Recall	0.7830	0.4110	0.5970
F1-score	0.6602	0.5049	0.5826
Accuracy	0.5970		

**Table 6 sensors-24-07815-t006:** Classification report for the HILL method.

Metrics	Clean Class	Stego Class	Average
Precision	0.6574	0.7318	0.6946
Recall	0.7830	0.5920	0.6875
F1-score	0.7147	0.6545	0.6846
Accuracy	0.6875		

**Table 7 sensors-24-07815-t007:** Classification report for the S-UNIWARD method.

Metrics	Clean Class	Stego Class	Average
Precision	0.7856	0.7925	0.7890
Recall	0.7950	0.7830	0.7890
F1-score	0.7903	0.7877	0.7890
Accuracy	0.7890		

**Table 8 sensors-24-07815-t008:** Classification report for the SSIS method.

Metrics	Clean Class	Stego Class	Average
Precision	0.9812	0.8195	0.9003
Recall	0.7830	0.9850	0.8840
F1-score	0.8710	0.8946	0.8828
Accuracy	0.8840		

**Table 9 sensors-24-07815-t009:** Comparison of SRNet performance in steganalysis against various methods.

Method	Original Model [1]					Fine-Tuned Model			
	PE	Prec	Rec	F1	Acc	PE	Prec	Rec	F1
WOW	0.0672	0.89	0.87	0.88	0.933	0.4020	0.57	0.78	0.66
HILL	0.1151	0.86	0.84	0.85	0.885	0.3125	0.66	0.78	0.71
S-UNIWARD	0.0705	0.88	0.86	0.87	0.929	0.2110	0.79	0.79	0.79
SSIS	0.2145	0.77	0.75	0.76	0.785	0.1160	0.98	0.98	0.98

Note: PE—Probability of Error, Prec—Precision, Rec—Recall, F1—F1-score, Acc—Accuracy.

## Data Availability

The datasets generated during and/or analyzed during the current study are available from the corresponding author on reasonable request.

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
