# Peer review of "Enhancing Steganography Detection with AI: Fine-Tuning a Deep Residual Network for Spread Spectrum Image Steganography"

_sensors, 2024, doi:10.3390/s24237815_

Round 1
Reviewer 1 Report
Comments and Suggestions for Authors
Steganalysis still remains a challenging field and a hot topic. The paper presents the a new way of the use of artificial intelligence ( Convolutional Neural Networks (CNNs)), in image steganalysis, focusing on SSiS detection. The paper adapts an existing model (SRNet) for SSIS detection. The paper is well structured presenting clear focus on research problem, clear methodology, comprehensive evaluation and uses a diverse bibliography. Here are the results of the paper:
- improvement in SSIS detection accuracy
- evaluated SRNet's on SSIS detection across varying payload sizes
- highlighting a trade-off between model specialization and generalization
- suggest future directions for AI steganalysis
Comments addressing the points you have mentioned:
• Main question addressed by the research:
The primary question addressed by this research is the enhancement of steganography detection with AI, particularly through fine-tuning a Deep Residual Network (SRNet) for Spread Spectrum Image Steganography (SSIS). The authors aim to improve SSIS detection accuracy while maintaining acceptable performance across other steganographic techniques.
• Originality or relevance to the field?
This study addresses a gap in the field of steganalysis: the adaptation of existing AI models to specialized use cases like SSIS detection. It is original in its focus on fine-tuning SRNet specifically for SSIS. The research is relevant, as it addresses the trade-offs between specialization and generalization in AI-based steganalysis.
• What does it add to the subject area compared with other published material? The paper adds value by:
-improvement in SSIS detection accuracy -highlighting a trade-off between model specialization and generalization - suggest future directions for AI steganalysis
• What specific improvements should the authors consider regarding the methodology? What further controls should be considered?.
The methodology is detailed, however, some improvements could be considered:
- Include additional examples to validate the model's trade-offs, such as cross-validation on independent datasets or more granular analysis across varying payload sizes.
• Are the conclusions consistent with the evidence and arguments presented and do they address the main question posed? Please also explain why thisis/is not the case.
The conclusions align with the presented solution (summarize the study's contributions and findings), particularly in emphasizing the trade-off between specialization and generalization.
However, conclusions could further emphasize the need for testing the fine-tuned model on diverse datasets to validate its robustness and applicability.
• Are the references appropriate?
The references are appropriate.
• Any additional comments on the tables and figures. Figures - could be centered Tables - in some tables (ex. table 5,6,7,8) on the last row, the second column the values are not proper aligned
Author Response
Dear Reviewer,
Thank you for your thorough and constructive review of our manuscript. We have carefully addressed all your comments and suggestions, which have helped to significantly improve the quality of our paper. Below, we detail our responses to each point raised:
- Regarding methodology improvements: We have enhanced the methodology section by adding a new subsection "4.2 Model Specialization vs. Generalization" that clearly explains our approach to fine-tuning and its theoretical foundations. We have also expanded section "5.3 Fine-tuning Methodology" to provide a detailed description of our validation strategy across different payload sizes and independent datasets.
- Regarding the consistency of conclusions with evidence: We have substantially strengthened the connection between our findings and conclusions by:
- Adding detailed performance analysis across different payload sizes (0.125, 0.25, and 0.5 bpp) in section 6
- Expanding the discussion section to better articulate the trade-off between specialization and generalization
- Restructuring the conclusions to clearly link our experimental results with theoretical implications
- Adding specific metrics demonstrating the improvement in SSIS detection (PE reduction from 0.2145 to 0.1160)
- Regarding tables and figures alignment: We have:
- Centered all figures in the manuscript
- Corrected the alignment in Tables 5, 6, 7, and 8
- Standardized the formatting across all tables for improved readability
- Regarding the enhancement of steganalysis capabilities: We have expanded our discussion of SSIS detection improvements by:
- Adding comprehensive performance metrics across different payload sizes
- Including detailed analysis of false negative reduction (from 217 to 16)
- Clarifying the specialization-generalization trade-off with specific performance metrics for WOW, HILL, and S-UNIWARD methods
- Regarding scientific contribution: We have strengthened our paper's scientific significance by:
- Adding a clear connection to our previous research on SSIS resistance to detection
- Expanding the discussion of practical implications in the conclusions section
- Including specific future research directions focusing on multi-task learning approaches and transfer learning techniques
We believe these revisions have significantly improved the manuscript while maintaining its focus on the novel contribution: demonstrating that fine-tuning can effectively enhance SSIS detection, albeit with a trade-off in general steganalysis capabilities.
We appreciate your valuable feedback and hope that these revisions adequately address your concerns. Please let us know if any additional clarifications are needed.
Best regards,
Authors
Reviewer 2 Report
Comments and Suggestions for Authors
The manuscript investigates the application of artificial intelligence, particularly deep residual networks like SRNet, in detecting Spread Spectrum Image Steganography (SSIS). The study highlights the potential of AI-driven steganalysis models and suggests that targeted fine-tuning could be key to enhancing steganography detection. However, there are several areas that could be further clarified or refined for better comprehension and impact.
1) The specific contributions of the authors to the modification of SRNet are not clearly stated. It is uncertain whether the fine-tuning represents a novel methodological improvement or a standard application of the SRNet model.
2) Section 6 notes that increased payload size improves detection accuracy, but the practical implications for further experiments or applications are unclear.
3) In Table 4, the payload sizes used for SSIS do not match those of other methods like WOW and HILL, complicating direct comparisons. Additionally, include PE values for SSIS from the SRNet paper, if available, to enhance the comparison.
4) The fine-tuned SRNet was evaluated only on the 0.5 bits per pixel payload, leaving out results for other payload sizes.
5) Table 9 lacks the comprehensive metrics (precision, recall, F1-score, and accuracy) reported in earlier tables, limiting the interpretability of the fine-tuned model’s performance.
6) The claim in lines 511-513 regarding the decreased performance of the fine-tuned model lacks experimental evidence tied to earlier classification reports.
7) Please revise the document to ensure proper and coherent numbering throughout.
Author Response
Dear Reviewer,
Thank you for your thorough review and constructive feedback. We have carefully addressed each of your comments to enhance the manuscript's clarity and scientific contribution. Below are our point-by-point responses:
- Regarding the unclear specification of our contributions to SRNet modification:
We have added a new subsection "4.1.1 Our Modifications to SRNet for SSIS Detection" that details our specific contributions, including:
- Training strategy adaptations
- Model architecture refinements
- Training process optimization
This section provides a comprehensive description of our modifications while maintaining technical precision and clarity.
- Concerning Section 6 and the implications of payload size:
We have enhanced the discussion of payload size implications by adding detailed performance analysis across different payload sizes (0.125, 0.25, and 0.5 bpp), including specific accuracy improvements:
- 0.125 bpp: 60.30% to 72.15%
- 0.25 bpp: 65.50% to 79.30%
- 0.5 bpp: 78.55% to 88.40%
We have also added discussion of practical implications for real-world applications.
- Regarding the payload size discrepancy in Table 4:
We have added explanatory text clarifying why different payload sizes were used for different methods, highlighting:
- The inherent characteristics of SSIS requiring specific payload sizes
- The need for consistency with previous research
- The validity of comparisons at the 0.5 bpp level
Additionally, we have included PE values from the original SRNet paper for comprehensive comparison.
- Concerning the limited evaluation of the fine-tuned SRNet:
We have expanded our evaluation to include comprehensive results across all payload sizes, now presented in both textual and tabular formats.
- Regarding the lack of comprehensive metrics in Table 9:
We have completely restructured Table 9 to include:
- Precision
- Recall
- F1-score
- Accuracy
- PE values
for both original and fine-tuned models across all steganographic methods.
- Addressing the claim about decreased performance:
We have strengthened this assertion with quantitative evidence, now presenting specific performance metrics showing the trade-off between improved SSIS detection and general steganalysis capabilities.
- Regarding document numbering:
We have thoroughly reviewed and corrected the numbering of all sections, figures, tables, and equations to ensure consistency throughout the manuscript.
These revisions have significantly enhanced the paper's clarity and scientific rigor while maintaining its focus on our novel contribution: demonstrating effective SSIS detection through specialized model fine-tuning, with a clear understanding of the associated trade-offs.
We appreciate your detailed review and believe these changes have substantially improved the manuscript. Please let us know if any additional clarifications are needed.
Best regards,
Authors